# Extending the toolbox for RNA biology with SegModTeX: a polymerase-driven method for site-specific and segmental labeling of RNA

Raphael Haslecker [1], Vincent V. Pham[1], David Glänzer[2], Christoph Kreutz [2], Theodore Kwaku Dayie [3] & Victoria M. D'Souza[1]✉

RNA performs a wide range of functions regulated by its structure, dynamics, and often post-transcriptional modifications. While NMR is the leading method for understanding RNA structure and dynamics, it is currently limited by the inability to reduce spectral crowding by efficient segmental labeling. Furthermore, because of the challenging nature of RNA chemistry, the tools being developed to introduce site-specific modifications are increasingly complex and laborious. Here we use a previously designed *Tgo* DNA polymerase mutant to present SegModTeX − a versatile, one-pot, copy-and-paste approach to address these challenges. By precise, stepwise construction of a diverse set of RNA molecules, we demonstrate the technique to be superior to RNA polymerase driven and ligation methods owing to its substantially high yield, fidelity, and selectivity. We also show the technique to be useful for incorporating some fluorescent- and a wide range of other probes, which significantly extends the toolbox of RNA biology in general.

RNA is positioned at a critical cross-section of biology: it not only disseminates genetic information, but also folds into structures to mediate many biological functions. This is due to its capacity to form numerous basepair types beyond the canonical Watson-Crick pairs, often providing the ability to sample multiple conformations that are critical to drive function[1–3]. Additionally, more and more nucleotide modifications at predefined sites are known to further expand the structural, mechanistic, and functional repertoire of RNA[4]. It is therefore important to understand RNA structures in their various conformations and in the correct chemical context to gain complete insights into its function and the biology it governs.

However, current RNA structural characterization methods suffer from three major drawbacks. The first is specific to NMR. While NMR provides structural data across the full folding landscape of a molecule, it fails for large RNAs due to signal overlap and broad linewidths.

Second, with current methods, we cannot easily construct functionally relevant chemical states of RNAs; for example, with existing enzymatic approaches, we cannot add at specific sites N6-methyl-adenosine (m6A), 5-methyl-cytosine (m5C), pseudouridine (Ψ), etc. Third, many biochemical techniques, such as FRET, have limited options on where the probe can be placed and hence are mostly confined to the termini, thus restricting complete characterization of the molecule[5].

One way to overcome both the size limitations of NMR and site-specific incorporation of modified nucleotides is to construct RNA segments sequentially, each of which can be manipulated for selective incorporation of isotopes (termed segmental labeling) and/or modified rNTPs and analogues[6]. There are three methods currently in use for segmental and site-specific labeling. First, one can synthetize individual RNA fragments and then ligate with T4 DNA or RNA ligase − Puglisi and coworkers used this approach to probe a 100 kDa HCV IRES

[1]Department of Molecular and Cellular Biology, Harvard University, Cambridge, MA 02138, USA. [2]Institute of Organic Chemistry and Center for Molecular Biosciences Innsbruck (CMBI), University of Innsbruck, Innrain 80/82, 6020 Innsbruck, Austria. [3]Department of Chemistry and Biochemistry, University of Maryland, College Park, MD 20782, USA. ✉e-mail: dsouza@g.harvard.edu

RNA by solution NMR[7,8]. Others, by extension, have added single isotopically labeled and modified bases at specific ligation sites using chemical or T7 RNA polymerase synthesis[9,10]. However, ligation itself has many shortcomings that severely limit its use including low yields in individual ligation and preparatory steps, sequence constraints at and around the ligation site, and minimum segment lengths[7]. Second, position-selective labeling of RNA (PLOR) uses T7 polymerase to extend RNA stepwise by using different unlabeled, labeled, or modified rNTP pools in individual steps[11]. T7 polymerase starts transcription de novo and cannot reengage with an RNA:DNA duplex. PLOR tries to circumvent this by keeping the RNA polymerase engaged with the duplex while the rNTP pools are washed on and off[12,13]. However, due to its technical complexity, use of saturating amounts of labeled rNTP pools per wash/step, and limited yield, few researchers have used this method so far.

Finally, a third approach involves chemical RNA synthesis using phosphoramidite nucleotide analogues. This method easily allows for selective incorporation of modified or isotopically labeled nucleotides[14]. However, chemical synthesis has a length constraint of ~70-nt for NMR spectroscopy. This limitation is imposed by the current stepwise coupling efficiency of ~98% in each step, which leads to overall low yields further diminished by side products during the deprotection steps[15,16]. Furthermore, many phosphoramidite versions of modified rNTPs are not commercially available and thus off limits to most labs[17]. Together, these methods have found limited application in overcoming the size and crowding limitations of NMR as only ~2% of deposited NMR structures in the RCSB database are greater than 70 nucleotides (as of April 2023) with the largest being 155-nt[8]. Out of

these, almost half required the use of segmental labeling via ligation to unambiguously assign resonances. Additionally, all deposited structures that incorporate site-specific RNA modifications either used short, chemically synthesized RNAs or RNAs extracted from biological material, both of which severely limit the type and composition of RNAs that can be studied.

While segmental incorporation of nucleotides in RNA is problematic, this is not the case for DNA. With primer extension, labeled or modified dNTP pools are easily incorporated into newly synthesized DNA. This is because DNA polymerases engage an existing primer template duplex and extend from the 3'-end of the primer, allowing the newly synthesized segment to be different from the primer depending on the dNTP mix provided[18]. On the other hand, virtually all RNA polymerases initiate transcription de novo and are not thermostable, making segmental additions or melting steps impossible. Nevertheless, Cozens et al. reported a mutant DNA polymerase (TGK) from *Thermophilus gorgonarius (Tgo)* with a unique capability to not only efficiently incorporate rNTPs, but importantly, also extend an RNA primer that is annealed to a DNA template[19]. Building on that work, we characterized and optimized the enzyme's fidelity, accuracy, versatility, and yield for extending RNA primers/segments. Here we present a method for 'Segmental labeling and site-specific Modifications by Template-directed eXtension' (SegModTeX, Fig. 1). We mainly use NMR as proof-of-principle, both to advance the field and due to its ease of detecting modifications and assessing sample quality. Furthermore, we describe a range of NTP analogues that are accepted by the polymerase, thus greatly expanding the toolbox for RNA biology.

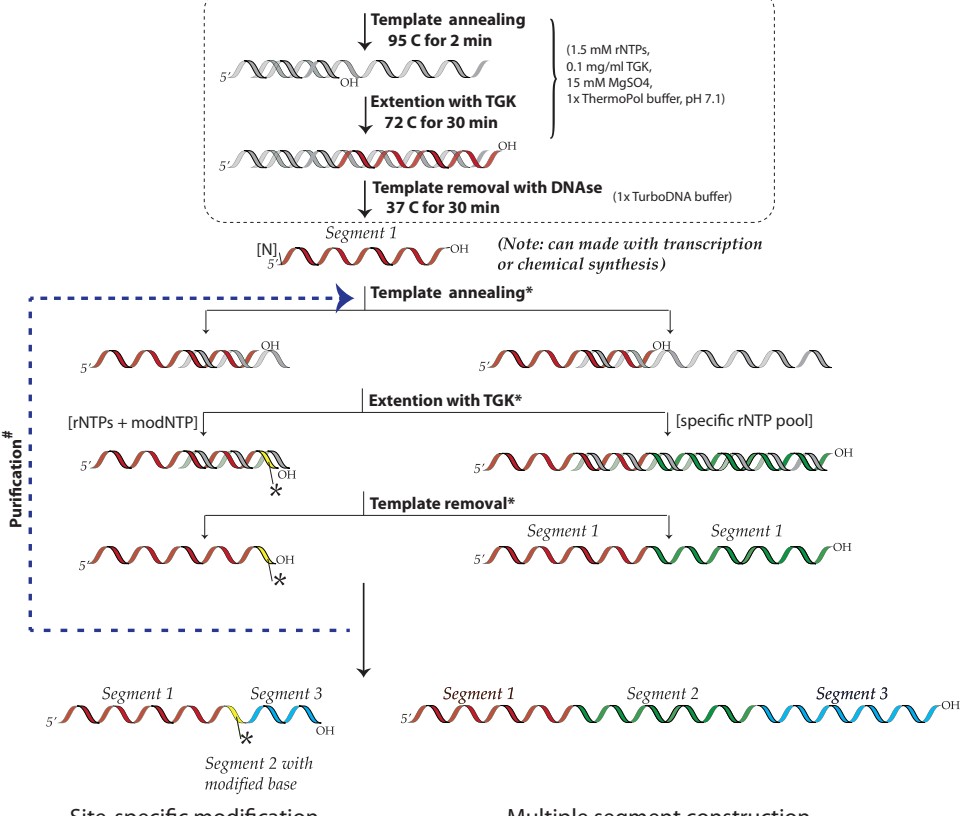

Site-specific modification — Multiple segment construction

**Fig. 1 | Schematic of the SegModTeX protocol.** The RNA segment 1 (red) is annealed to the reverse complementary template 1 (grey) encoding the second segment and a region complementary to the 3'-end of segment 1 (~20nt). Upon extension of segments 2 (yellow and green) using a pool containing the desired modified (left) or isotopically labeled (right) rNTPs, the template is removed and the extended RNA of segment 1 + 2 can undergo another round of extension (segment 3, blue). If the final product requires a non-G start, annealing a DNA primer to a template can undergo SegModTex to generate segment 1 (upper dotted box).

## Results

### Extending and modifying RNAs with high efficiency and fidelity by SegModTeX

For high quality and easy purification of TGK polymerase, we used an N-terminal GST-tag followed by a PreScission protease cleavage site upstream of the polymerase sequence. We obtained yields of 25 mg of protein per liter of *Escherichia coli* culture, which is comparable to that obtained for T7 polymerase[20] (Supplementary Fig. 1a).

We tested the purified TGK polymerase for properties that would make it conducive for segmental extension and labeling. First, we confirmed the enzyme's gain-of-function for incorporating rNTPs by extending a diverse set of RNA segments of varying lengths on DNA templates also of varying lengths. Segment 1 (seg1) of the various lengths and compositions were either made by T7 polymerase or by chemical synthesis, including HBV-epsilon (HBV-ε) ($25_{seg1}$), tRNA$^{lys3}$ ($20_{seg1}$), and 7SK snRNA ($41_{seg1}$). Each seg1 was annealed to a ssDNA template with a complementary 3'-end (-20-nt, $T_m$: -65 °C) and the sequence of seg2 encoded at the 5'-end. TGK was then used to extend the various seg1 to produce seg1–seg2: HBV-ε (+1 to $26_{seg1-seg2}$), tRNA$^{lys3}$ (+12 to $32_{seg1-seg2}$), and 7SK snRNA (+29 to $70_{seg1-seg2}$).

We rigorously tested all relevant parameters, including concentrations of templates, segments, rNTPs, enzyme, and divalent ions and additionally optimized buffer, pH, annealing conditions, temperature, and time for extension. Unlike T7 polymerase, all SegMod-TeX reactions were robust and went to completion under the same optimized reaction conditions of 0.1 mM template:seg1, 1.5 mM rNTPs, 0.1 mg/ml TGK, 15 mM MgSO$_4$, 1 x ThermoPol buffer (pH: 7.1), at 65 °C (short seg1) or 72 °C for <90 min. Under these conditions, we find the turnover of the various seg1 to seg1–seg2 approaches 100% as evidenced by the complete absence of seg1 in the enzymatically extended lanes and quantification of reaction yields (Fig. 2a and Supplementary Table 1). Overall, we find SegModTeX to be robust across a wide range of temperatures, incubation times, and pH values. First, SegModTeX extends even at room temperature; however, the minimum required temperature for complete extension is 55 °C, presumably due to barriers to melting secondary structures in the template. The upper bound is dependent on the annealing temperature of template:seg1 (Supplementary Fig. 1b). Second, while RNA can undergo hydrolysis at high temperatures, especially in the presence of Mg$^{2+}$, we only observed degradation for incubation times that far exceeded the

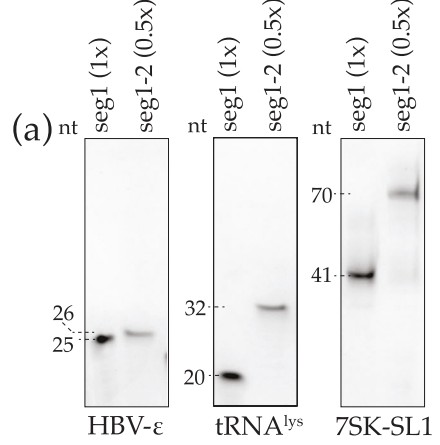

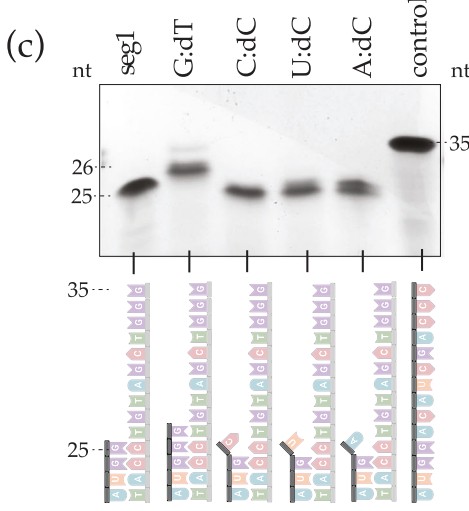

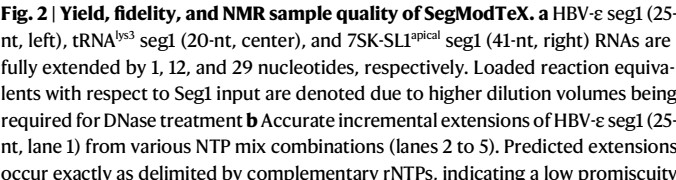

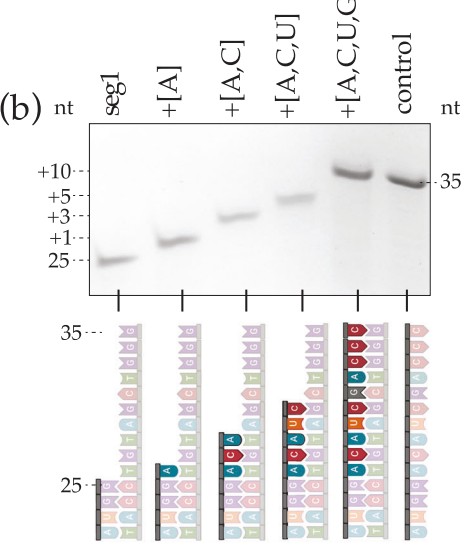

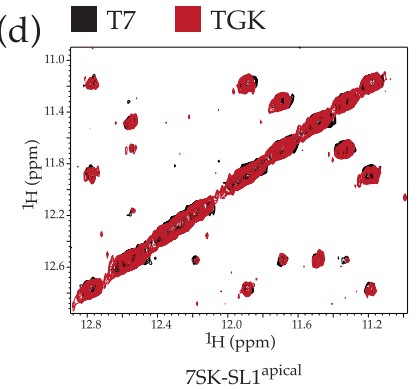

**Fig. 2 | Yield, fidelity, and NMR sample quality of SegModTeX. a** HBV-ε seg1 (25-nt, left), tRNA$^{lys3}$ seg1 (20-nt, center), and 7SK-SL1$^{apical}$ seg1 (41-nt, right) RNAs are fully extended by 1, 12, and 29 nucleotides, respectively. Loaded reaction equivalents with respect to Seg1 input are denoted due to higher dilution volumes being required for DNase treatment **b** Accurate incremental extensions of HBV-ε seg1 (25-nt, lane 1) from various NTP mix combinations (lanes 2 to 5). Predicted extensions occur exactly as delimited by complementary rNTPs, indicating a low promiscuity of the polymerase. **c** Lack of HBV-ε seg1 (25-nt, lane 1) extension with mismatched 3'-end base pairs (C:dC, U:dC, A:dC, lanes 3-5, respectively) indicates high fidelity. Additionally, a 26-nt HBV-ε seg1 with a G:dT mismatch extends inefficiently (lane 2), demonstrating a strong preference for canonical Watson-Crick base pairing. **d** 2D-NOESY overlays of 7SK-SL1$^{apical}$ (56-nt) constructed by T7 polymerase (black) and by SegModTeX extension of a 28-nt seg1 (red). Source data are provided as a Source Data file.

optimum range (Supplementary Fig. 1c). Finally, there is no observable difference in activity between pH 5-7 and only a 20% decrease at pH 8, after which, as expected, there is a dramatic loss in yield (Supplementary Fig. 1d).

Next, as TGK can use an expanded range of rNTP analogues as substrates, we tested if misincorporation of complementary bases occurs, which could lead to increased heterogeneity of the RNA constructs and render the method inadequate for segmental labeling. Therefore, we extended HBV-ε seg1 ($25_{seg1}$) using different combinations of rNTPs (A, AC, ACU, and ACUG) and checked if the enzyme stalls at the predicted site (+1, +3, +5, +10, respectively), or if the polymerase extends beyond that by using the wrong rNTPs. Remarkably, SegModTeX reactions proceed only up to the length expected and have a hard-stop when the complementary NTP is missing, highlighting its high fidelity, a feature that is a prerequisite for segmental labeling (Fig. 2b).

Furthermore, for segmental labeling, it is essential that accurate segment ends are used. In fact, in canonical ligation methods, terminal ribozymes are added to the desired sequence because T7 polymerase has a propensity to add non-templated bases at the end[21–23]. This is of special concern with long RNA segments where purification at single-nucleotide resolution is not feasible. Thus, we tested whether TGK retains its DNA polymerase property to discriminate and not use mismatched base pairs as substrates, which would allow for production of only accurate segment junctions[24]. We extended HBV-ε seg1 ($25_{seg1}$) with different versions of mismatched 3′-end base pairs (G:dT, C:dC, U:dC, A:dC) and performed extension assays. Mismatched segments are not extended (Fig. 2c); in fact, even the G:dT pair, which has been shown to be efficiently extended by Taq polymerase shows almost no detectable extension[24].

Finally, to confirm the high yield and fidelity of SegModTeX, we compared NMR spectra of a 56-nt 7SK-SL1$^{apical}$ RNA made by T7 polymerase to that made by TGK (28-nt$_{seg1}$ to 56-nt$_{seg1-seg2}$)[25]. 7SK-SL1$^{apical}$ is especially enriched with bulges and non-Watson base pairs which are readily discernable by NMR. Our comparative analysis (Fig. 2d) shows that the sample made by segmental labeling is indistinguishable from that made by T7, emphasizing the high fidelity and accuracy of SegModTeX.

## SegModTeX allows for rapid NMR assignments of large multidomain RNAs

Since current segmental labeling techniques are inefficient and laborious, RNA assignments by NMR largely rely on atom- or nucleotide-specific labeling. Atom-specific labeling strategies only allow for assignments of up to 70-nt with ease. Furthermore, deuteration, wherein proton positions are substituted for deuterons, leads to loss of structural information at those sites, which is especially critical for solving the structures of loops and bulges. Nucleotide-specific labeling strategies have been used to investigate RNAs over 100-nt but are a resource- and time-consuming process. It requires the parallel assignments from numerous nucleotide specifically labeled samples; for example, assignments of the MLV-Ψ packaging signal (SL-BCD) required four specifically deuterated samples, four $^{15}$N/$^{13}$C labeled samples, each of which required 3D and 4D datasets[26]. For comparative analysis, we used SegModTeX to make the three-domain SL-BCD construct wherein the B domain (seg1, 35-nt) was left protonated while domains C and D were extended using deuterated rNTPs, rendering these domains invisible to NMR (seg1–seg2, 101-nt) (Fig. 3). Thus, use of such segmental labeling would only require three domain-specific protonated samples, each of which can be used to fully assign the respective domain, thereby significantly reducing the time required to solve the structure. Furthermore, this strategy can be combined with atom-specific deuteration at ribose moieties to significantly reduce the overlap, and further aid in assignments of full domains within large RNAs[27].

## SegModTeX allows for multiple extensions without segment length or sequence constraints

Assignments of domains in the middle of RNAs, for example domain C in the SL-BCD above would traditionally require two or more steps of differentially labeled segments. Thus, we tested if multiple segments of RNA can be efficiently added using SegModTeX. For this we used two variations of segment designs to produce a 324-nt 7SK snRNA. In the first, we designed a multi-step extension protocol to only visualize guanosines and adenosines from residues 181 to 253, while completely deuterating the segments on either end (7SK snRNA$_{180/253}$) (Fig. 4a). In the second, we wanted to visualize only the five adenosines present in regions between 149 and 178, while completely deuterating all other

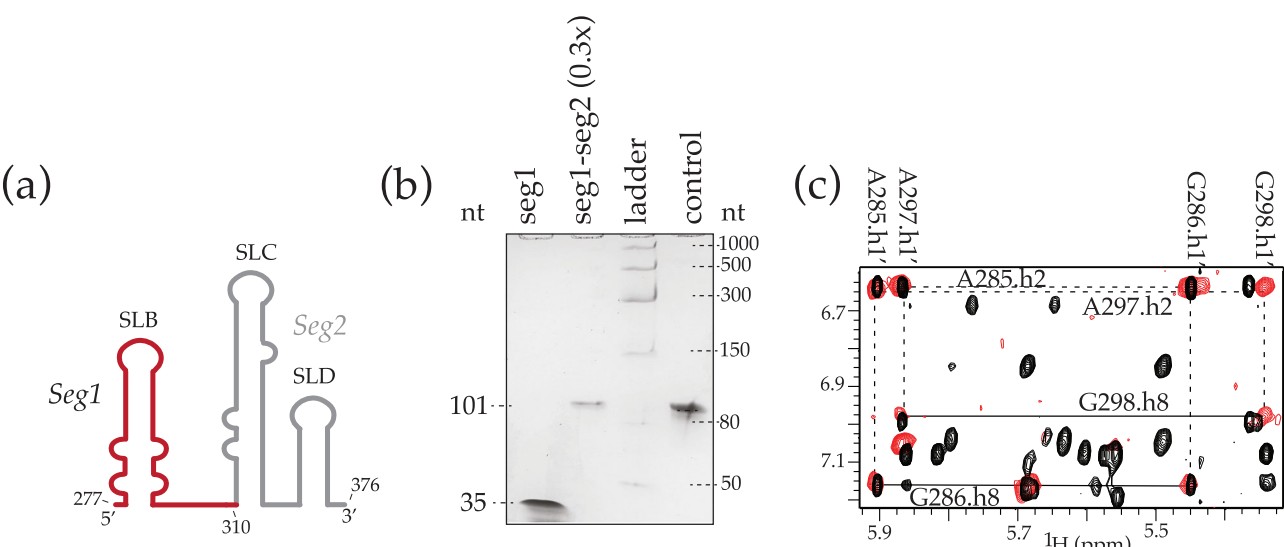

**Fig. 3 | Labeling large RNA segments for crowded spectra (MLV).** a Schematic of the segmental labeling of MLV-BCD. Seg1 (red) is fully protonated, whereas seg2 (grey) is fully deuterated. b Extension of seg1 (lane 1) to seg1-2 (lane 2). The loaded reaction equivalent with respect to Seg1 input are denoted due to higher dilution volumes being required for DNase treatment. c 2D-NOESY overlay of segmentally deuterated MLV-BCD RNA (red) and fully protonated MLV-BCD (black). The vastly simplified spectra allows for easy peak assignments in the context of the full RNA. Source data are provided as a Source Data file.

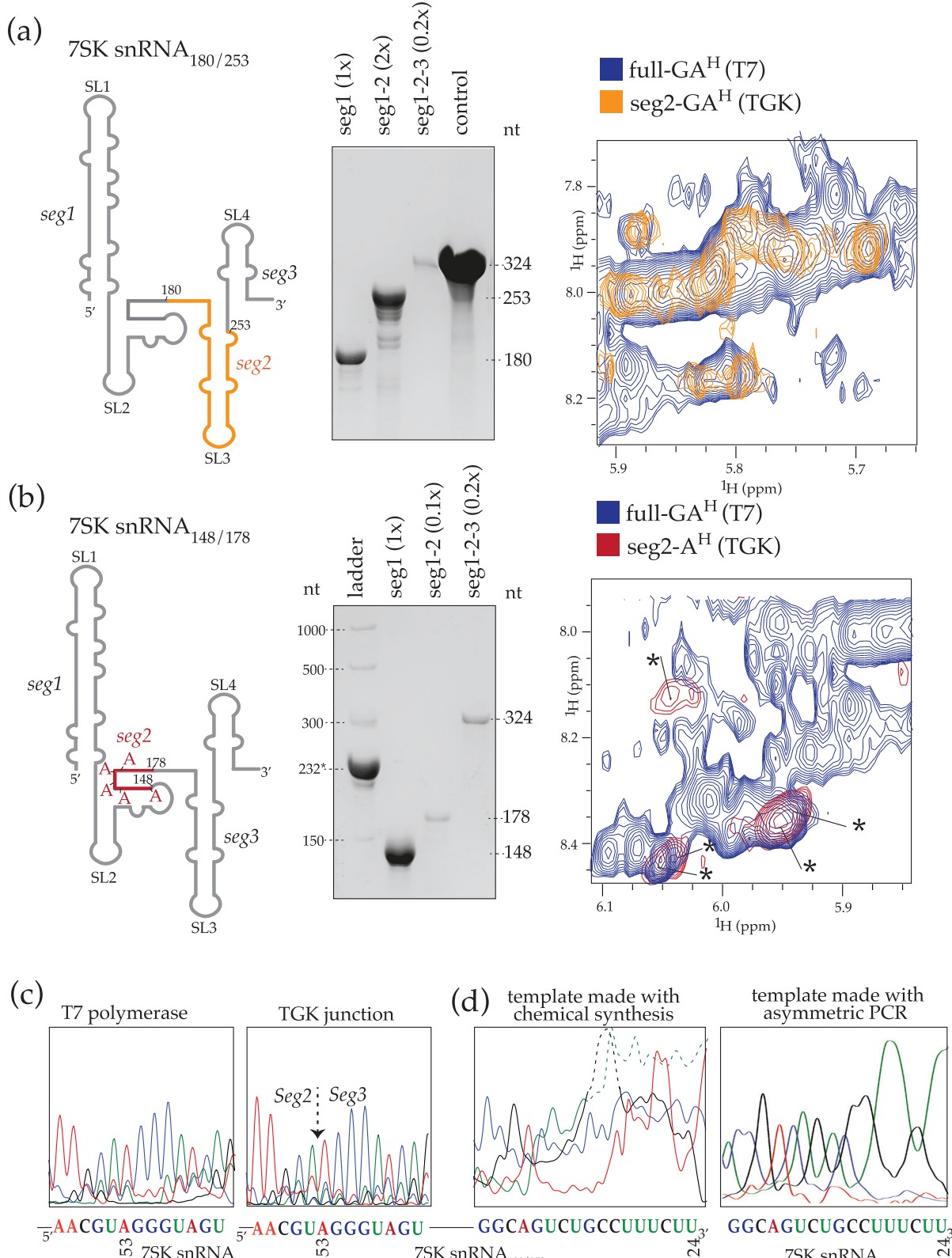

**Fig. 4 | Multi-segment labeling of large 7SK snRNAs.** Left: Schematics of segmental labeling of 7SK snRNA. Seg1 and 3 (black) are fully deuterated in both. In **a** only C and U are fully deuterated, while all G and A are protonated in seg2 (orange), whereas in **b**, seg2 (red) has G, C, and U fully deuterated and only five A's protonated. Center: Stepwise extensions of RNA segments 1 to 3 [lanes 1-3 in (**a**) and 2–4 in (**b**)]. Loaded reaction equivalents with respect to Seg1 input are denoted. Right: Overlay of 2D-NOESY spectra of a T7-transcribed G and A protonated sample

(blue) and segmentally labeled samples [orange for (**a**) and red for (**b**)].
**c** Comparative RT-seq of 7SK snRNA NMR samples made by canonical T7 and the SegModTex schema in (**a**) match at the junction of seg2 and seg3. **d** Comparative RT-seq of the SegModTex 7SK snRNA NMR samples using chemically synthesized ssDNA template (left) and asymmetric PCR derived template (right) show degenerate ends and correct ends, respectively. The dashed portion of the plots are scaled down to fit the displayed area. Source data are provided as a Source Data file.

bases (7SK snRNA$_{148/178}$) (Fig. 4b). After extending seg1 to seg1–seg2 and subsequent DNase digestion, we repeated SegModTeX using new ssDNA templates to add seg3. Comparison with a G and A protonated, full-length 7SK snRNA made by T7 polymerase shows the individually identifiable proton shifts of the specified nucleotides in both samples; for example, five expected adenosines are visible in 7SK snRNA$_{148/178}$ that align reasonably well with the resonances of the control sample. Such drastic reduction of spectral complexity should easily allow for structural characterizations of specific domains and/or nucleotides in large RNAs (Fig. 4a, b).

To confirm that non-complementary base pairs are not extended (see Fig. 2c), we allowed for incomplete extensions of seq1–seg2 during the extension of 7SK snRNA$_{180/253}$ (Fig. 4a). While shorter fragments would naturally lack complementarity to the next template and thus be excluded from further extensions, mismatched fragments that still anneal to the template would cause inaccurate junctions. Sequencing of 7SK snRNA$_{180/253}$ shows that the junction made in the 7SK snRNA$_{180/253}$ sample is comparable to that region made by non-segmented T7 polymerase transcription, confirming the above observation that SegModTeX does not extend inaccurate or mismatched ends (Fig. 4c). On the other hand, we find that the 3′-ends of the 7SK snRNA$_{180/253}$ seg3 has significant heterogeneity. Given the high fidelity of the enzyme, we reasoned this heterogeneity arose from varying template sizes produced by chemical DNA synthesis. To circumvent this heterogeneity problem, we tested the use of asymmetric PCR to construct a homogenous ssDNA template. We synthesized the whole 324-nt 7SK snRNA template sequence, which was used for the 7SK snRNA$_{148/178}$ synthesis above[28]. As expected, sequencing results showed that the 3′ end was accurately extended, highlighting the ability of SegModTeX to construct RNA segments of any length, with the heterogeneity arising from the input templates of varying lengths.

As for the specification of 5′ ends, one current limitation when making any RNA construct, or in this case, preparation of seg1, is that T7 polymerase only initiates transcription from at least one guanosine with T7 class III promoter or, with lower efficiency, adenosine with a class II φ2.5 promoter[29,30]. Since SegModTeX extension of RNA is a novel function of a mutant DNA polymerase, we surmised it could also add RNA onto DNA segments without sequence constraints. Hence, we tested the extension of a short DNA segment annealed onto a longer ssDNA template encoding HBV-ε. The RNA sample obtained after DNase digestion is identical to that made by T7 polymerase (Fig. 5a). Therefore, SegModTeX can be used to synthesize the first RNA segment as well, making its design independent of any sequence constraints.

## Site-specific incorporation of modified NTPs by SegModTeX

Many RNAs are modified in cells to control function. The most prevalent modifications are N$^6$-methyladenosine (m$^6$A), N$^1$-methyladenosine (m$^1$A), inosine (I), 2′-O-methylation (2′-OMe), 5-hydroxymethylcytosine (hm$^5$C), 5-methylcytosine (m$^5$C), and pseudouridine (Ψ)[31]. Cozen et al. showed that the latter two could substitute for Cs and Us[19]. Given that SegModTeX can be used for multi-segment extension, we wanted to test if we can site-specifically introduce these and other biologically relevant modifications.

We first tested incorporation of Ψ at position 27 in tRNA$^{lys3}$ and m$^6$A at position 26 (HBV-genome: 1907) in the distal HBV-ε stem-loop[32,33]. Both Ψ and m$^6$A are widely used modifications and occur naturally in these constructs. In the first step, we used the same template:seg1 setup as before (Fig. 2), except that the single base extension in HBV-ε was carried out in the presence of m$^6$ATP and then extended to completion in the second step (Fig. 5b). Similarly, a 12-nt seg2 extension for tRNA$^{lys3}$ was carried out in the presence of ΨTP, where U is specified only once in the sequence. A subsequent extension with regular rNTPs was used to complete the tRNA$^{lys3}$ to 76-nt (Fig. 5c). NMR assignments confirmed incorporation of the m$^6$A at the

correct position 26 in HBV-ε through the presence of an upfield shifted resonance (~ 2.8 ppm)[3], which is absent in the sample made by T7 polymerase without an m$^6$A modification (Fig. 5b). Similarly, NMR assignments of tRNA$^{lys3}$, confirmed the incorporation of Ψ at position 27 by the expected disappearance of the H5 resonance of U$_{27}$, which is seen in the samples made by T7 polymerase (Fig. 5c). In both cases, no other resonance of m$^6$A or Ψ were observed and the spectra showed that the RNAs folded similar to their unmodified counterparts, indicating that the incorporation occurs accurately and only at the specified site. Finally, to further advance the NMR field, we introduced 2-$^{19}$F-ATP in the HBV-ε construct at the same nucleotide position as m$^6$A described above. As seen in Fig. 5d, the 1D spectra confirms the incorporation of a single 2-$^{19}$F-Adenine in the RNA. We also show that 2-$^{19}$F-2-$^{13}$C-ATP can be incorporated in large RNAs, which are of special interest to NMR spectroscopists, for example in 7SK snRNA$_{148/178}$ (Supplementary Fig. 1f).

We also tested other biologically relevant base modified NTPs including inosine, 5-methyl-C, and 2-thio-U. To test these, we made various HBV-ε templates that encoded each modified base only at a single specified site (position 29 of 35 for modified UTPs and position 28 of 32 for all other modified NTPs). Extension stalls at expected lengths in the absence of the specified NTPs but goes to completion with modified NTPs: The HBV-ε (25$_{seg1}$) extends by 7 nucleotides (for CTP-, GTP-, and ATP-analogues (Fig. 6a, c)) or 10 nucleotides (for UTP-analogues, Fig. 6b).

Since SegModTeX readily incorporated these base modified NTPs, we wanted to test if bulkier functionalized NTP analogues that can be used for various biochemical techniques would also be incorporated. Thus, we tested: (i) Alexa-Fluor-555-aha-dCTP and Fluorescein-12-dUTP for fluorescence labeling; (ii) 5-Aminoallyl-CTP, 5-Aminoallyl-UTP, and 5-Ethynyl-UTP for crosslinking; (iii) biotin-16-UTP for affinity biology; (iv) 5-Bromo-UTP, 5-Iodo-UTP, and 5-Iodo-CTP for halogeno labeling. As seen in Fig. 6a–c, all the base modified NTPs, with the exception of Alexa-Fluor-555-aha-dCTP, are incorporated and fully extended. Alexa-Fluor-555-aha-dCTP is also incorporated but leads to termination without further extension (Fig. 6d). Given that Fluorescein-12 UTP and Alexa-Fluor-555-aha-dCTP are similar in size, it is unclear what leads to this phenomenon with one possibility being the difference in linker length between the base and the fluorophore.

Next, we wanted to test if modifications on the sugar moieties would be incorporated. We tested both biologically relevant 2′-O-methyl-NTPs, as well as dNTPs, ddNTPs, and 2′ (or 3′) linked-Alexa Fluor™ 647 ATP, -BODIPY FL ATP, and -MANT-GTP. As expected, dNTPs are fully extended while ddNTPs cause chain termination at the specified site after incorporation. Surprisingly, even a small moiety like a methyl at the 2′ position is not conducive for extension (Fig. 6a–c); however, the NMR data shows that termination occurs only after incorporation of the 2′-O-methyl (Fig. 6e). Accordingly, two bands of varying intensities are observed for reactions containing a mix of fluorophores that are linked to either the 2′ or 3′ positions: one stalled due to the incorporation of the 2′-modified NTP, and one fully extended, but without the fluorophore present, presumably due to its expulsion during polymerization and formation of the 3′ phosphodiester linkage (Fig. 6d). Overall, these results show that SegModTeX is not conducive for labeling of the 2′ position of RNAs at internal sites but can be used successfully for labeling the last residue of RNA constructs.

## Discussion

RNA biology is a rapidly advancing field; however, the construction of RNA samples with native modifications remains challenging. Furthermore, NMR, one of the main methods for structural analysis accounting for 35% of deposited RNA structures[17], is severely limited due to the complexity of assigning spectra without the ability to isotopically label individual segments. Finally, biochemical methods such as FRET, RNA-crosslinking, etc., are confined to end-labeling or

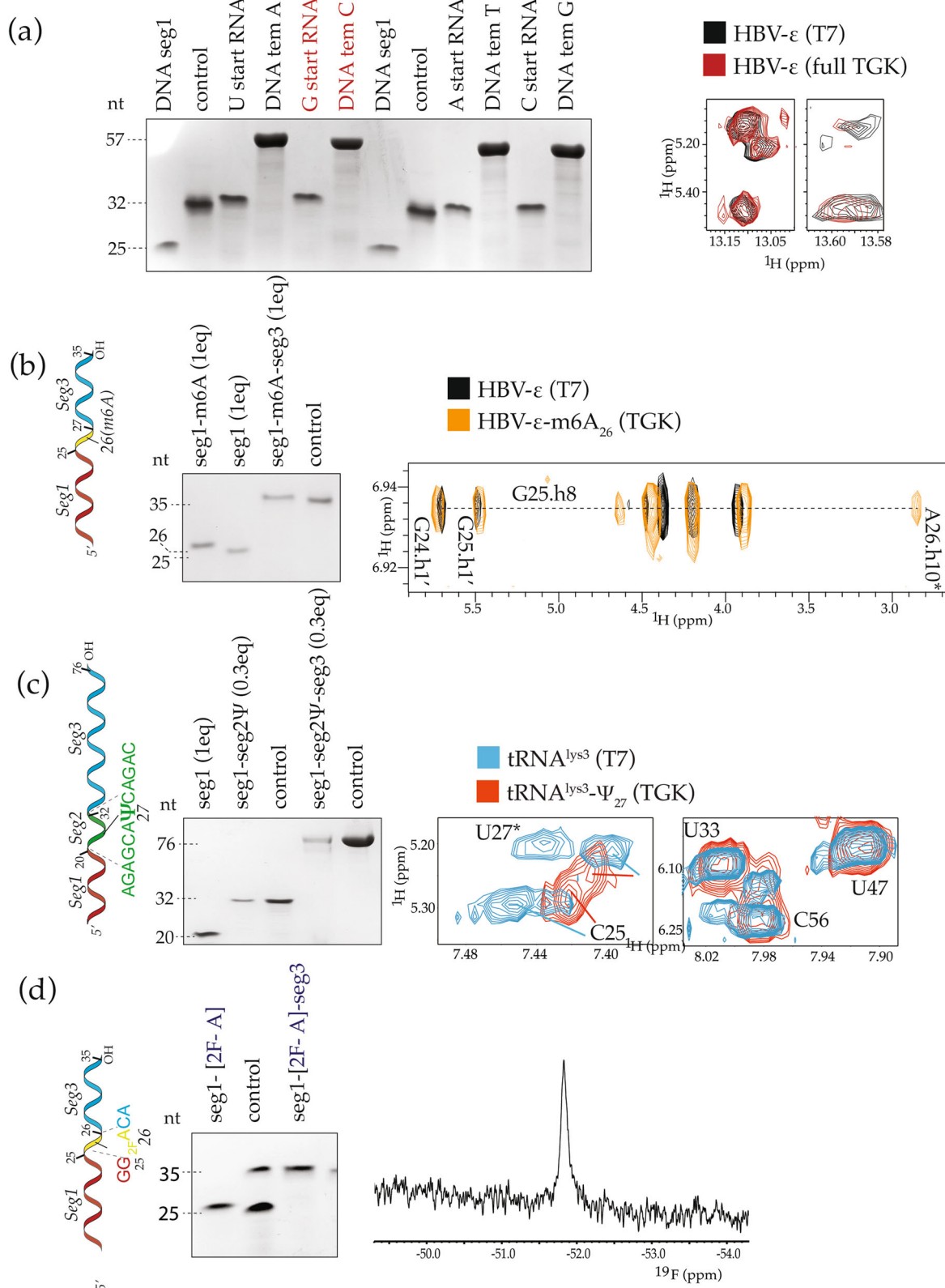

cumbersome ligation techniques to achieve site-specific incorporation. In this paper we showcase the use of a mutant *Tgo* DNA polymerase for 'Segmental labeling and site-specific Modifications by Template-directed eXtension' (SegModTeX) – a one-stop protocol to address all these limitations.

First, unlike ligations, the extension reaction goes to completion, which addresses the biggest hurdle of segmental labeling. Thus, insofar

as the product is recovered efficiently, the yield can approach 100%. Even the stringent method of gel electrophoresis followed by electro-elution produces yields of around 80% of the input. This in principle allows for multiple rounds of extensions. Second, although the enzyme has been mutated to accommodate rNTPs, it has not resulted in a loss of complementary base recognition, maintaining the high fidelity of DNA polymerases. This directly benefits junction homogeneity because

**Fig. 5 | Multi-segment extension with SegModTeX. a** Left: DNA-primed RNA synthesis via SegModTex. DNA seg1 (25-nt, lane 1) is extended with U, G (red), A, and C starts (lanes 4, 6, 10, 12, respectively). Upon DNase digestion, 32-nt RNA products remain (lanes 3, 5, 9, 11, respectively). Right: Overlay of 2D-NOESY spectra of DNA-primed SegModTeX-derived (red) and T7-derived (black) HBV-εstem 1, **b** Left: Schematic and PAGE of HBV-ε seg1 (lane 1) extension by a single m⁶A segment (lane 2) followed by a 9-nt segment (lane 3). Loaded reaction equivalents with respect to Seg1 input are denoted. Right: Overlay of 2D-NOESY spectra of a SegModTex-derived m⁶A-modified (orange) with an unmodified T7-transcribed sample (black). The asterisk denotes the cross-peak of the N⁶-methyl protons and the H8 proton of residue G25 **c** Left: Schematic and PAGE of tRNA$^{lys3}$ seg1 (lane 1) extended by a 12-nt seg2 containing Ψ at position 27 (lane 2) followed by a 44-nt seg3 (lane 4) using SegModTeX. Loaded reaction equivalents with respect to Seg1 input are denoted. Right: Overlay of 2D-NOESY spectra of the Ψ-modified tRNA$^{lys3}$ (red) and T7-transcribed fully protonated tRNA$^{lys3}$ (blue). The asterisk denotes the H5 proton of U27 in the non-modified tRNA$^{lys3}$, which is absent in the Ψ-modified tRNA$^{lys3}$. **d** Left: Similar to (**b**) but extended by a single 2-¹⁹F-A segment (lane 1) and then the 9-nt segment (lane 3) Right: 1D-NMR spectra shows the incorporation of the 2-¹⁹F-A into only one position in HBV-ε RNA. The corresponding full spectrum showing a larger 20 ppm chemical shift region is overlaid with spectrum with a smaller 5 ppm chemical shift region to highlight the fact that only one 2-¹⁹F atom is incorporated. Source data are provided as a Source Data file.

mismatched 3′-ends are consequently selected against for extension; therefore, the end product only contains the desired sequences. Similarly, unlike RNA polymerases, no RNA self-templating occurs[34], leading to homogeneous 3′-ends to the extent that the ssDNA template is accurate. In fact, use of asymmetric PCR to make high quality ssDNA template renders the process independent of any sequence length constraints that arise from restrictions of chemical synthesis.

Third, since all extensions require the same optimized conditions, no further optimization for reaction conditions or special sequence considerations are required. Moreover, compared to even a single-step ligation protocol, which involves separate preparation of the two segments that are appropriately protected or primed for ligation followed by splinting and addition of ligase, SegModTeX allows for a one-step reaction to achieve segment joining. Fourth, compared to T7 polymerase-driven RNA production, this method has a few additional advantages, in that no base-type restrictions exist for the initiating nucleotide and the use of high temperature can resolve any potential terminators in the RNA. On the other hand, the high temperatures used in SegModTeX may preclude studies of RNA that require co-transcriptional folding. Furthermore, the retention of DNA-primed extension by the enzyme allows seg1 to have no minimum length requirement because the RNA segment alone does not have to be long enough to anneal. Finally, the salient feature of SegModTeX is the ability to use multi-step extensions to site-specifically incorporate a plethora of modified NTPs at any desired position(s) without sequence constraints around the modified site(s). For example, one could specifically modify or label any cytosine, even if it is flanked by other cytosines, via single nucleotide extensions.

This versatility and ease of SegModTeX has the potential to transform RNA structural biology. First, this technique can be used to build more sophisticated tools for existing methods. For example, currently RNA FRET is mostly limited to labels at the two ends, whereas SegModTeX permits easy incorporation of fluorescent labels at least Fluorescein-12 at any position on an RNA, thus granting access to the whole molecule. Second, we see similar potential for selective halogeno-labeling for x-ray crystallography, crosslinking, or affinity tags. Finally, SegModTeX has the potential to transform structure determination by NMR. For instance, structures of smaller domains can be solved within its larger context due to the drastic reduction of spectral crowding. This, along with the current progress in assignment prediction software can allow for a rapid workflow for characterizing large RNAs currently outside the scope of solution NMR[35]. Thus, we anticipate that this technology (together with others) will enable NMR spectroscopy to play a central role in illuminating the structure, dynamics, and function RNAs in vitro and in cells.

## Methods
### TGK polymerase
The plasmid for TGK-polymerase expression was cloned to encode an N-terminal Glutathione-S-transferase, a PreScission protease cleavage site, and the polymerase insert between the BamHI and NotI restriction sites of a pGEX-6P-3 vector. Plasmid DNA was transformed into One Shot™ BL21 Star™ cells and grown to 2 L from overnight cultures.

Subsequent IPTG induction at ~O.D. 0.7 was followed by 3–4 h growth. Cells were concentrated by centrifugation, resuspended in 40 ml lysis buffer (25 mM Tris pH 7.5, 300 mM NaCl, 1 mM EDTA, 2 mM DTT), and sonicated on ice with six 1-min cycles at 60–70% duty and 5-6 output control, with 1 min cool down between each pulse. Lysate was incubated with washed Pierce™ Glutathione Agarose slurry from Thermo-Fisher (cat. #16101) in lysis buffer for 2 h at 4 °C. Subsequently, the beads were washed with 750 ml lysis buffer and cleaved overnight in 10 ml of elution/storage buffer (0.01 M Tris-HCl pH 7.6, 0.1 M KCl, 0.1 mM EDTA, 1 mM DTT, 0.1 % v/v Triton X-100, 5 % glycerol) with 60 μl aliquot of PreScission protease (13 mg/ml made inhouse). After elution from the column, TGK was stored at 4 °C for short-term or diluted with one volume glycerol at −20 °C for long-term. Typical yield is around 25 mg per liter of culture. The purity after elution is shown in Supplementary Fig. 1a.

### DNA template design
The ssDNA templates contain a 3′end reverse complementary to the 3′ end of the RNA segment for their hybridization. The length should translate to a melting temperature ($T_m$) of ~65 °C, though lower melting temperatures are possible if the reaction is run at lower temperatures (see below). Longer hybridization regions do not affect SegModTeX. As the template is single-stranded, it is advised to avoid long self-complementarity at the 3′ends to avoid the template serving as primer for the TGK polymerase in an off-target reaction, however, as the reaction occurs at 72 °C, we have not encountered this problem in any tested construct thus far. Moreover, if self-templating is observed, non-extendable ssDNAs with di-deoxy or 3′-monophosphate ends can be used.

### DNA template preparation
For SegModTeX templates shorter than 100-nt, ssDNA was ordered from Integrated DNA Technologies (IDT) and if necessary, purified on a 6% polyacrylamide 25% formamide sequencing gel. SegModTeX templates longer than 100-nt were prepared using asymmetric PCR as described in[28], extracted via a 2-step ammonium acetate (2.5 M) and 70% ethanol precipitation, and subsequently washed and concentrated with $H_2O$ in Amicon® Ultra-15 Centrifugal Filter Units.

Similarly, for control samples made by T7 polymerase, shorter templates were ordered from IDT, with 2′-O-methylated 5′-ends[36]. For the longer 7SK snRNA samples, plasmids containing the T7 promoter, insert, and SmaI recognition sequence were cloned in between the EcoRI and BamHI restriction sites of a pUC19 vector. Plasmid DNA was prepared for PCR amplification from a 2 mL overnight culture of NEB 5α Competent *Escherichia coli* (C29871) transformed with the plasmid. Subsequently, the precise template sequences were amplified using primers ordered from IDT and EconoTaq® PLUS 2X Master Mix. PCR products were extracted via a 2-step ammonium acetate (2.5 M) and 70% ethanol precipitation.

### RNA seg1 preparation
RNA seg1 for the various constructs were synthesized by in vitro transcription using T7 RNA polymerase. Chemically synthesized RNAs

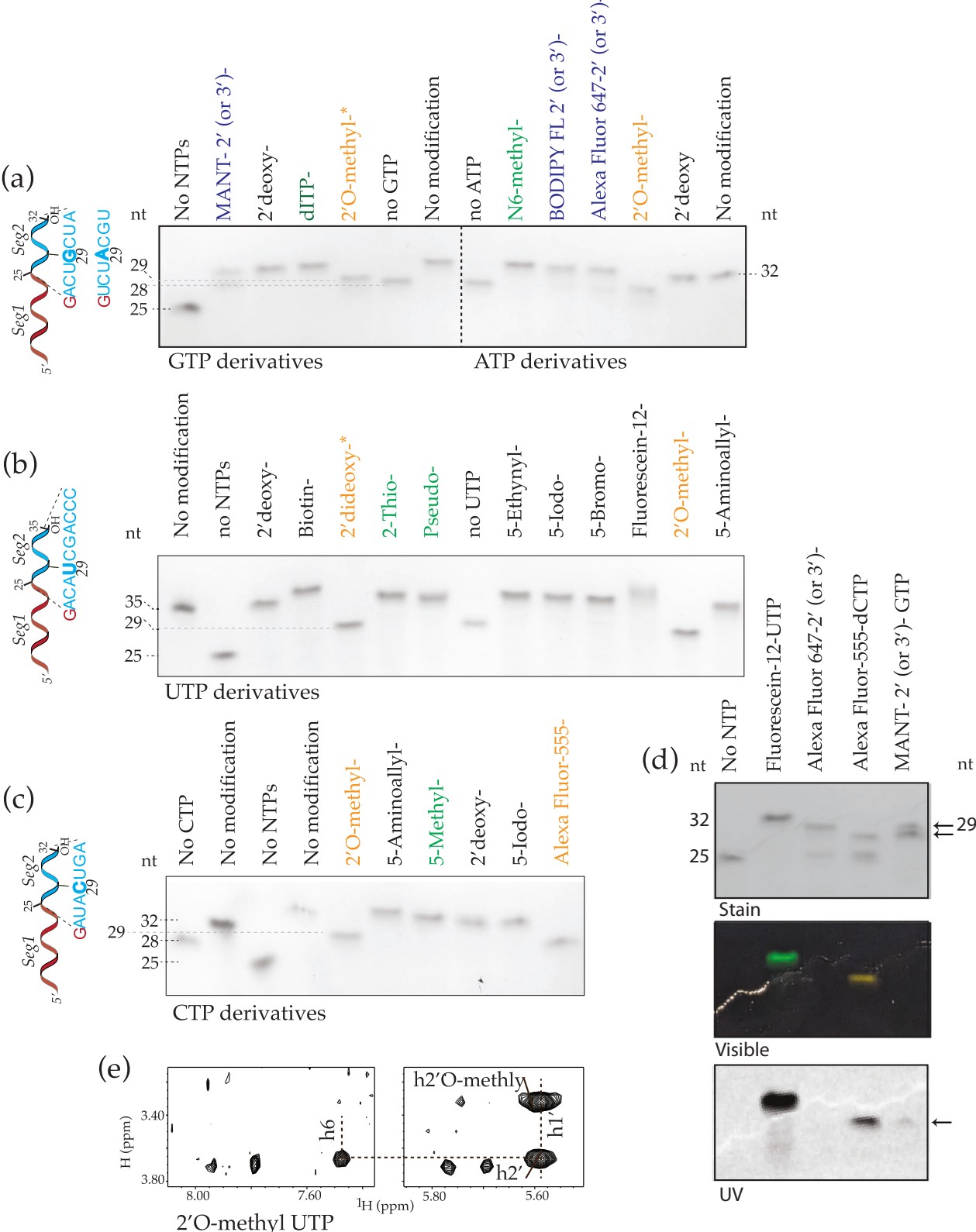

**Fig. 6 | Site-specific incorporation of modified NTPs by SegModTeX.** Extensions of 25-nt HBV-ε seg1 using a set of three regular rNTPs and a fourth modified NTP. Schematic (left) and PAGE (right) are shown of ATP/GTP **a** UTP **b** and CTP **c** analogues. Biologically relevant NTP analogues (green lanes) are incorporated and fully extended. Orange lanes indicate analogues that are incorporated but are then stalled. Blue lanes show fluorescent analogues that either are incorporated but stall or fully extend by eliminating the probe moiety. **d** Visible light and UV imaging of fluorescent analogue extensions. Fluorescein-12 (lane 2) is incorporated and fully extended, whereas Alexa-Fluor-647-ATP, Alexa Fluor-555-aha, and MANT-GTP (lanes 3–5, respectively) lead to two products. One fraction stalls (arrows) and exhibits color in visible light (middle) and UV (bottom), while the other fraction fully extends but shows no evidence of incorporation in light and UV readings. The Alexa-Fluor-647-ATP's stalled band is too faint for detection (visible). **e** 2D-NOESY spectra of HBV-ε seg1 (25-nt) shows successful incorporation of 2'O-methyl UTP before stalling. Source data are provided as a Source Data file.

were also tested for constructs with RNA seg1 smaller than 26-nt (ordered from IDT). Resuspended RNA segments from in vitro transcription and ethanol precipitation or chemical synthesis were directly used for SegModTeX. For most reactions, the RNA was PAGE purified before every round of SegModTeX to avoid any carryover of rNTPs from previous reactions which SegModTeX might mistakenly incorporate into the new segment as well. However, we found that for smaller RNA constructs (<70-nt), ammonium acetate – ethanol precipitation successfully removes rNTPs from the previous step.

## SegModTeX reaction conditions

**Reaction mix**. All SegModTeX reactions were conducted according to the following optimized protocol: 1 x ThermoPol buffer (NEB), 15 mM $MgSO_4$, 1.5 mM rNTPs, 0.1 mM RNA seg:ssDNA template, and TGK polymerase to a final concentration of -0.1 mg/ml were mixed at 4 °C. For large RNAs, SUPERase•In™ RNase Inhibitor was added according to the manufacturer's specifications.

**NTPs**. The NTP mix contained the individual bases at concentrations according to their demand in the extension reaction. For rare or expensive NTPs, ratios as low as 15-fold excess per base incorporation were successfully tested. Higher concentrations of NTPs require special considerations for the DNase treatment step (see below).

**Temperature**. The reaction mixture was incubated at either 65 °C (for RNA segments with a calculated $T_m$ of the template hybridization sequence of less than 63 °C) or else 72 °C. If experimental design requires very short RNA:DNA hybridization regions (<-15-nt), extension can be started at lower temperatures to allow for the addition of initial bases so as to increase the $T_m$, after which the temperature can be increased to allow for fast completion.

**Time**. The SegModTeX reactions were incubated for 20 min (extensions of fewer than 30-nt) to up to 90 min. Extended incubation time of up to 90 min can be beneficial for long extensions or highly structured segments but did not impact RNA integrity in our experiments. As TGK is a thermostable polymerase, heat-denaturation, either at the beginning or in between the incubation, akin to PCR cycles, was successfully tried and can be used for reaction optimization but was not required for any RNA sample shown. Moreover, we found that increasing the polymerase concentration was an easier and effective way to achieve extension completion.

**DNA removal and RNA purification**. Template DNA was digested with Turbo DNase according to the manufacturer protocol (37 °C, 15 min). It is important to note that incomplete DNA digestion has a detrimental impact on data quality due to the difficulty in separating it from the RNA in subsequent steps. Hence, for the digestion, special consideration should be given to the maximum allowed DNA concentration as specified in the Turbo DNase protocol as high template and NTP concentrations are inhibitory. This also limited the amount of SegModTeX product added in polyacrylamide gel wells due to volume constraints. In most cases, three-fold dilution of the reaction mix was necessary and sufficient to ensure complete digestion. To stop the reaction, EDTA was added equimolar to the $Mg^{2+}$ concentration and denatured at 95 °C for 2 min.

For purification without PAGE, ½ volumes 7.5 M ammonium acetate were added, the samples vortexed for 10 s, placed on ice for 10 min, and centrifuged at 15,000 g for 20 min at 4 °C. This step is necessary to remove the enzymes in the sample, which might interfere with applications downstream. In a new tube, the supernatant and 4 volumes ethanol were added, mixed, and precipitated for 2 h at −80 °C. The centrifuged pellet was washed twice with 80% cold ethanol, dried at room temperature for 6 h, and resuspended in $H_2O$. The sample was then ready for another round of SegModTeX.

For PAGE purification, the digestion reaction was precipitated using 0.3 M sodium acetate in 4 volumes ethanol at −80 °C and subsequently resuspended in RNA gel-loading buffer (50% formamide, 25 mM EDTA) and purified using urea-PAGE with 25% formamide. The choice of purification method depends on the subsequent use of the RNA. As the extension of RNA in SegModTeX is essentially 100%, size-based separation of the RNA product is not necessary in general. However, as the sample is in a mix of salts, proteins, NTPs, dNMPs and DNA oligos, precipitation or filter-based methods might be insufficient for highly sensitive applications, such as NMR.

## Yield quantification

For yield quantifications by centrifugal filtration, the reactions were washed six times with 5 M Guanidinium•HCL, 1 M Tris pH 8.0 at 12.6 krpm for 10 min at room temperature. The concentrates were then diluted in water (1:50) and ODs were measured using a Nanodrop 2000 Spectrophotometer after blanking with a similarly diluted wash solution. For an approximate yield quantification of the long exposure and pH titration experiments (Supplementary Fig. 1c, d), NIH ImageJ 1.53 k was used to measure the intensity of the individual product size bands, normalized to lane 1, respectively.

## NMR data acquisition and resonance assignment

RNA samples were suspended in the appropriate buffer (MLV: 5 mM Tris•HCl, 10 mM NaCl, pH: 7.0, 37 °C; 7SK snRNA: 5 mM Tris•HCl, pH: 7.0, 25 °C; 7SK-SL1[apical]: 10 mM potassium phosphate, 70 mM NaCl, and 0.1 mM EDTA, pH 5.2; HBV: $D_2O$, pH: 7.0, 25 °C; tRNA: 10 mM Tris•HCl, 10 mM NaCl, 7 mM $MgCl_2$, pH: 7.5 buffer, 37 °C). All NMR experiments were acquired in 5 mm Shigemi tubes with Bruker 700 and 800 MHz instruments containing cryogenic probes. Spectra for observing non-exchangeable protons were collected in 100% $D_2O$ and for exchangeable protons in 90% $H_2O$/10% $D_2O$. All $^{19}F$/$^1H$ spectra were collected at 298 K using a Bruker 600 MHz Avance III spectrometer equipped with BBFO (broad band with $^{19}F$ observe and $^1H$ decoupling) probes. Spectra were taken with carrier at −51.8 ppm. All data was analyzed using NMRDraw v11.1, NMRviewJ 8.0.3, and NMRFx Analyst v11.2.4-c.

## Sequencing

The DNA oligo (5'TAATACGACTCACTATAGGGTCTCTTGTTCATG AGTCATGG3') was 5' phosphorylated with T4 PNK (NEB, #M0201S) and ligated onto the 3' ends of PAGE purified SegModTeX and T7 constructed 7SK snRNA NMR samples with T4 RNA ligase 1 (NEB, #M0437M) according to the manufacturer's protocols. Reverse transcription was performed with a short (25-nt) primer (5'TCCATGACT-CATGAACAAGAGACCC3') using GoScript™ Reverse Transcriptase (Promega A5003). PCR amplification of the RT product was done with EconoTaq® PLUS (LGC Biosearch Technologies #300331) with the RT primer and primers complementary to the 5' start of 7SK snRNA. The agarose-gel purified PCR products were sent for Sanger sequencing.

## Statistics and reproducibility

Representative gel images from Fig. 2a–c, Fig. 4a, b, to Fig. 5 b–d are from n = 3 replicates whereas images from Fig. 3b, Fig. 5a, to Fig. 6a–d from n = 2 replicates.

## Reporting summary

Further information on research design is available in the Nature Portfolio Reporting Summary linked to this article.

# Data availability

All data presented in the manuscript are available from the corresponding author upon request. Source data and sequences for the DNA and RNA oligoes used are provided with this paper. Source data are provided with this paper.

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

## Acknowledgements

This work was supported by HHMI grant 55108516 (to V.M.D.) and NIH grant U54 AI50470 (formerly U54 GM103297) and U54 AI170660 (both to V.M.D. and T.K.D.). We thank Samuel Carlson for providing PreScission protease.

## Author contributions

R.H. conceived, designed, and optimized the experiments. R.H. and V.V.P. prepared the samples and performed the NMR experiments. R.H. performed the fidelity, NTP analogue, multi-step, and reviewer experiments. R.H. and V.M.D. analyzed and interpreted the data and wrote the manuscript. D.G. and C.K. synthesized and provided the 2-19F-2-13C-ATP and C.K., T.K.D., and V.V.P. edited the manuscript.

## Competing interests

The authors declare no competing interests.

## Additional information

Victoria M. D'Souza.

