## [Peer Review File · Nature Communications]

Extending the toolbox for RNA biology with SegModTeX: A polymerase-driven method for site-specific and segmental labeling of RNAREVIEWER COMMENTS

Reviewer #1 (Remarks to the Author):

Haslecker et al., introduces a new method for segmental labeling of RNA, termed Segmental labeling and site-specific Modifications by Template-directed eXtension, or SegModTeX. This method employs a mutant (Tgk) *Thermophilus gorgonarius* (Tgo) DNA polymerase that can incorporate ribonucleotides and extend RNA primers, to drive step-wise, ssDNA-templated, segmental RNA extension and position-specific RNA labeling. The authors demonstrate that this versatile thermostable enzyme exhibits exceptional processivity, high accuracy and fidelity, and can accept a wide range of chemically modified nucleotides. Using mostly NMR, the authors observed comparable quality, spectra, and structures for SegModTeX-derived RNA samples with those generated by T7 RNA polymerase. Finally, the authors also report some limitations of the enzyme and method, such as reduced ability to extend beyond 2'-omethyl NTPs and certain Alexa dye-conjugated NTPs. As site-specific labeling of long RNAs is a chief technical barrier for most biophysical (such as single-molecule FRET) and structural studies (NMR, X-ray crystallography) of complex RNAs, this easy-to-use, broadly applicable method, built on a robust, precise, and processive DNA polymerase platform, holds outstanding promise as the go-to method for segmental RNA labeling. The study is meticulously designed, rigorously controlled, and technically sound. The manuscript is succinctly written and clearly illustrated. I recommend this work for publication in Nature Communications with enthusiasm and provide a few minor comments, questions, and suggestions below, which may help further clarify and improve the manuscript.

Specific comments:

1. The SegModTeX condition contains 15 mM Mg²⁺ and incubates RNA samples at 72°C and for up to 90 min. As high temperature and high Mg²⁺ conditions are known to elicit in-line cleavage of certain RNAs, the authors should probably discuss this potential concern, and describe whether unscheduled cleavage was occasionally observed, such as Fig. 4a, lane 2?
2. Is it known what are the temperature and pH tolerances of this enzyme? If the requirement for higher temp is a limitation of this enzyme, is there future prospect to develop a mesophilic version? In the method (pg 26) the authors briefly mention that the enzyme "starting to extend RNAs even at room temperature". Can the authors provide more information, such as whether the enzyme is reasonably active at 37 °C?
3. On the flip side, the higher temperature also has an advantage over T7 RNAP in that stable RNA structures such as intrinsic terminators are reduced, and that the higher temp may provide more thermodynamic motivation to facilitate the folding of complex RNA structures.
4. Fig. 1, after DNase digestion of the DNA-RNA chimera, is it known what is left on the 5'-end of the RNA segment, phosphate or 5'-OH?
5. Fig. 4d, the dashed lines should be defined.
6. Fig. 6. I think having the sequence within this figure would be helpful for the readers to better correlate observed stalls with their exact positions and sequence contexts. Also, it would be helpful to indicate the exact lengths of the stalled products.
7. Fig. 6. The authors suggest that 2'-O-methyl NTPs cause chain termination like the dideoxy-NTPs. Do the authors have evidence that the 2'-O-methyl NTP, or Alexa555 NTP themselves are incorporated? For instance, the authors can examine if the RNA become fluorescent in the case of Alexa555 NTP.
8. Fig. 6. Do the authors see any patterns of which modified NTPs are better than others in terms being accepted by Tgo DNAP? Based on the available structural data referenced in Cozens et al. 2012, is that a potential rationale for the observed selectivity? For instance, would the authors predict that all bulky 2'-OH adducts would be rejected by Tgo? Related to this, is there a known reason that the Alexa 647-ATP is incorporated okay but Alexa 555-CTP is not?
9. Pg 27, para 2, line 9. "insure" should read "ensure".

Reviewer #2 (Remarks to the Author):

This is a wonderful contribution from the D'Souza, Kreutz, and Kwaku Dayie labs which addresses a long-standing technological need in the field – the ability to robustly site-specifically and segmentally label RNAs with ease high yields and fidelity. A good number of example applications are presented. I recommend publication after the authors address the following minor comments and suggestions:

1. Figure 3 (right) why not also show the non-deuterated sample for comparison as was done in figure 4. Also label the different panels with letters.
2. The authors should show the NMR spectrum of the ^{19}F fluorinated seg2 of the 7SK snRNA148/178 to establish the utility of the method for NMR studies.
3. I would strongly encourage the authors to provide 1D ^1H spectra for all the prepared RNA samples and overlay the labeled RNAs with their unlabeled counterparts. In addition, for every main figure showing a zoomed in region of the spectrum, the corresponding full spectrum showing the entire chemical shift region should also be provided. These can be provided in supporting material.
4. Although not required for publication, I was disappointed not to see $^{13}\text{C}/^{15}\text{N}$ labeling – the authors should comment on the feasibility of such labeling given their wide utility in the NMR field.

Reviewer #3 (Remarks to the Author):

The manuscript by Haslecker and co-workers presents an intelligent method for the labelling of nucleic acids based on an DNA polymerase mutant identified in the Holliger group a few years ago. The method allows to both introduce modified nucleotides at specific positions and to label with, for example, stable isotopes entire segment of the RNA. It has, potentially a broad range of applications in experiments looking at structure, dynamics and interactions of RNA. The work characterizes the activity of the polymerase in single and multistep reactions, which explores the range of possibilities and defines the experimental parameters.

The paper is overall well written and the figures are clear, although at times the information is not copious. The topic is relevant and the results interesting, this is in principle an elegant exploration of a very useful method.

Below are a number of points that needs to be addressed prior to publication.

Major point:

- 1- While the specificity of the reaction is generally clear, the yield, which is important to establish the potential of this strategy, is not really reported. The authors should report the final amount of RNA products per ml of reaction, for the different reactions they have performed.

Minor points:

- 1- Purification of the TGK polymerase (page 5). Please describe in detail and show gels, chromatograms etc. This is important information to be able to use this method. This could be in the Supplemental materials.
- 2- I would add a panel to Figures 1 where the actual experiment is represented in a cartoon form and the standard parameters are reported. This I think would help link the concept to the experimental conditions.

3- Page 5, one but last line. The bands of Seg1-seg2 in 2a are weak. It seems unlikely you one a seg1 band which is significantly weaker than seg1-seg2. Based on what I see on this gel, one can say seg1-seg2 is the only major species, but I am not sure how the authors can state with confidence that the turnover is 100%. How is this measured.

4- Page 6 last line of the second paragraph and figure 2c. It seem to me G:dT runs as an extended oligo, i.e. as a 26mer.

5- Figure 2 – The legend for Figure 2d is missing

6- Figure 3, please add the spectrum of the fully labelled MLV, as a comparison.

7- Page 8, end of the first paragraph and Figure 4b. One of the adenine resonances, labelled with an asterisk, is not superimposing. Could you please explain and correct the statement in the paper as required.

8- Figure 5c, right. The spectrum of the AC and psi RNAs show what is probably noise at the frequency of U27. Could you please confirm and comment?

9- It would be good to see repeats for the key reactions in the Supplemental methods.

REVIEWER COMMENTS

Reviewer #1 (Remarks to the Author):

Haslecker et al., introduces a new method for segmental labeling of RNA, termed Segmental labeling and site-specific Modifications by Template-directed eXtension, or SegModTeX. This method employs a mutant (TGK) *Thermophilus gorgonarius* (Tgo) DNA polymerase that can incorporate ribonucleotides and extend RNA primers, to drive step-wise, ssDNA-templated, segmental RNA extension and position-specific RNA labeling. The authors demonstrate that this versatile thermostable enzyme exhibits exceptional processivity, high accuracy and fidelity, and can accept a wide range of chemically modified nucleotides. Using mostly NMR, the authors observed comparable quality, spectra, and structures for SegModTeX-derived RNA samples with those generated by T7 RNA polymerase. Finally, the authors also report some limitations of the enzyme and method, such as reduced ability to extend beyond 2'-omethyl NTPs and certain Alexa dye-conjugated NTPs. As site-specific labeling of long RNAs is a chief technical barrier for most biophysical (such as single-molecule FRET) and structural studies (NMR, X-ray crystallography) of complex RNAs, this easy-to-use, broadly applicable method, built on a robust, precise, and processive DNA polymerase platform, holds outstanding promise as the go-to method for segmental RNA labeling. The study is meticulously designed, rigorously controlled, and technically sound. The manuscript is succinctly written and clearly illustrated. I recommend this work for publication in Nature Communications with enthusiasm and provide a few minor comments, questions, and suggestions below, which may help further clarify and improve the manuscript.

1. The SegModTeX condition contains 15 mM Mg²⁺ and incubates RNA samples at 72°C and for up to 90 min. As high temperature and high Mg²⁺ conditions are known to elicit in-line cleavage of certain RNAs, the authors should probably discuss this potential concern, and describe whether unscheduled cleavage was occasionally observed, such as Fig. 4a, lane 2?

We thank the reviewer for bringing up this important point. We did not observe any significant cleavage in the time required for completion of the extension. To test for this, we have now included an experiment with increasing incubation times (Supplementary Fig. 1a), which shows significant degradation at more than twice the maximum recommended incubation time. We have added this observation to the main document (Lines 127-143):

“Unlike T7 polymerase, all SegModTeX reactions were robust and went to completion under the same optimized reaction conditions of 0.1 mM template:seg1, 1.5 mM rNTPs, 0.1 mg/ml TGK, 15 mM MgSO₄, 1x ThermoPol buffer (pH: 7.1), at 65°C (short seg1) or 72°C for < 90 min. Under these conditions, we find the turnover of the various seg1 to seg1-seg2 approaches 100% as evidenced by the complete absence of seg1 in the enzymatically extended lanes and quantification of reaction yields (Fig. 2a and Supplementary Table 1). Overall, we find SegModTeX to be robust across a wide range of temperatures, incubation times, and pH values. First, SegModTeX extends even at room temperature; however, the minimum required temperature for complete extension is 55°C, presumably due to barriers to melting secondary structures in the template. The upper bound is dependent on the annealing temperature of template:seg1 (Supplementary Fig. 1b). Second, while RNA can undergo hydrolysis at high temperatures, especially in the presence of Mg²⁺, we only observed degradation for incubation times that far exceeded the optimum range (Supplementary Fig. 1c). Finally, there is no observable difference in activity between pH 5-7 and only a 20% decrease at pH 8, after which, as expected, there is a dramatic loss in yield (Supplementary Fig. 1d).”

2. Is it known what are the temperature and pH tolerances of this enzyme? If the requirement for

higher temp is a limitation of this enzyme, is there future prospect to develop a mesophilic version? In the method (pg 26) the authors briefly mention that the enzyme “starting to extend RNAs even at room temperature”. Can the authors provide more information, such as whether the enzyme is reasonably active at 37 °C?

We have now included a gel that demonstrates the effect of temperature on extension capabilities of SegModTeX. As mentioned, the enzyme is active even at 37°C, but does not complete the extension, potentially due to barriers for resolving secondary structures in the template. We have included this data in supplemental figure 2 and updated the written description.

3. On the flip side, the higher temperature also has an advantage over T7 RNAP in that stable RNA structures such as intrinsic terminators are reduced, and that the higher temp may provide more thermodynamic motivation to facilitate the folding of complex RNA structures.

Yes, we have added a statement about the advantage of high temperature in terms of termination reduction (line 341). However, there is no thermodynamic advantage for folding of complex RNA structures because the extended RNA is involved in RNA:DNA hybrids until DNase digestion. We have also now added a caveat to this SegModTeX design in that it cannot be used for RNA structures that are dependent on co-transcriptional folding (line 343).

4. Fig. 1, after DNase digestion of the DNA-RNA chimera, is it known what is left on the 5'-end of the RNA segment, phosphate or 5'-OH?

From the NMR experiment we know that the DNA digestion is complete. We assume that Turbo DNase leaves a 5' phosphorylated end on the RNA, as it does on a DNA strand (see manufacturer website).

5. Fig. 4d, the dashed lines should be defined.

Done. The description is provided in the legend for Fig. 4d.

6. Fig. 6. I think having the sequence within this figure would be helpful for the readers to better correlate observed stalls with their exact positions and sequence contexts. Also, it would be helpful to indicate the exact lengths of the stalled products.

Done. We added schematics for each part of the figure showing the sequence of the extension and the site where the NTP derivatives are incorporated. We have also added dashed lines to denote the stalled lengths.

7. Fig. 6. The authors suggest that 2'-O-methyl NTPs cause chain termination like the dideoxy-NTPs. Do the authors have evidence that the 2'-O-methyl NTP, or Alexa555 NTP themselves are incorporated? For instance, the authors can examine if the RNA become fluorescent in the case of Alexa555 NTP.

To address this question, we have now included NMR data in Fig. 6e showing that chain termination occurs after incorporation of 2'-O-methyl UTP. We have also included gels that show

the incorporation of Alexa-555 CTP before termination in comparison to 12-Fluorescein UTP control, which extends fully after incorporation of the modified nucleotide (Fig. 6d and line 288). Both data indicate that termination occurs after incorporation; thus, this method can be used to label the 3' end of an RNA construct.

8. Fig. 6. Do the authors see any patterns of which modified NTPs are better than others in terms being accepted by Tgo DNAP? Based on the available structural data referenced in Cozens et al. 2012, is that a potential rationale for the observed selectivity? For instance, would the authors predict that all bulky 2'-OH adducts would be rejected by Tgo? Related to this, is there a known reason that the Alexa 647-ATP is incorporated okay but Alexa 555-CTP is not?

We thank the reviewer for asking details about incorporation of modifications.

In terms of the modifications that are on the base, we find that all tested modifications are readily incorporated and extended with the exception of Alexa-555-aha-dCTP, which is incorporated but then aborted. We do not have an explanation for this given that Fluorescein-12, which is also a base modification on a UTP and of a similar size is incorporated and extended. It is possible that the differences in the linker between the base and the dye is the determining factor. We show this data both in Fig. 6d and Supplementary Fig. 4, where both dyes, one aborted and one fully extended, are clearly visible on the gel before staining.

In terms of modifications that are on the sugar moiety, we find that even a methyl in the 2'-O-position is not tolerated for extension. As mentioned above, we show NMR data in Fig. 6e, where we clearly see the incorporation of the 2'-O-methyl. Given this, we reinvestigated incorporation of the fluorophores that are available as mixtures of 2' or 3' modifications, for example MANT-GTP and Alexa-647-ATP. As our figure shows, these always gave two bands, an aborted and an extended version. Our newer data with UV and visible light imaging (see Fig. 6d) shows that while the aborted band has incorporated the fluorophore, the extended one has no fluorophore. Our best explanation is that the 2'-modified version is incorporated and then aborted just like 2'-O-methyl, whereas the 3'-modification is incorporated but then lost as the phosphodiester linkage for the next nucleotide is made at that position.

We have reworded this section and added this new observation in the manuscript (lines 294-307).

9. Pg 27, para 2, line 9. “insure” should read “ensure”.

Done.

Reviewer #2 (Remarks to the Author):

This is a wonderful contribution from the D'Souza, Kreutz, and Kwaku Dayie labs which addresses a long-standing technological need in the field – the ability to robustly site-specifically and segmentally label RNAs with ease high yields and fidelity. A good number of example applications are presented. I recommend publication after the authors address the following minor comments and suggestions:

1. Figure 3 (right) why not also show the non-deuterated sample for comparison as was done in figure 4. Also label the different panels with letters.

Done.

2. The authors should show the NMR spectrum of the ^{19}F fluorinated seg2 of the 7SK snRNA_{148/178} to establish the utility of the method for NMR studies.

For simplicity, we have rearranged our presentation of the data and showed the incorporation of ^{219}F -ATP in the same construct as the $m^6\text{A}$. The 7SK snRNA_{148/178} does not allow for a good comparison because it's a non-native shortened version of the RNA and we do not have a good characterization of that sample.

3. I would strongly encourage the authors to provide 1D ^1H spectra for all the prepared RNA samples and overlay the labeled RNAs with their unlabeled counterparts. In addition, for every main figure showing a zoomed in region of the spectrum, the corresponding full spectrum showing the entire chemical shift region should also be provided. These can be provided in supporting material.

We have added three supplementary figures (Supplementary Fig. 3) showing the entire chemical shift region for the aromatic to H1' correlations. One important point to note is that, as expected, while the majority of the peaks match with the control samples, the regions around the introduced modifications show perturbations.

For the HBV-epsilon- $m^6\text{A}$ sample, the modification introduces significant changes in the structure that is important for its biology ([doi.org:2120485119](https://doi.org/2120485119)). This is one of the main areas of research in the lab and thus, while we understand the importance of showing the data, we worry that this will be unnecessarily confusing and potentially also take away from our next publication.

Similarly, the HBV-epsilon- ^{219}F -A sample also introduces significant chemical shift changes in the residues near the fluorinated base. Both the D'Souza lab and the Dayie lab have observed this phenomenon independently in different samples. We are in the process of understanding the basis of this phenomenon for a future publication.

Therefore, we believe that the HBV-epsilon data is beyond of the scope of this paper.

4. Although not required for publication, I was disappointed not to see $^{13}\text{C}/^{15}\text{N}$ labeling – the authors should comment on the feasibility of such labeling given their wide utility in the NMR field.

We have included a gel in Supplementary Fig. 1e showing the incorporation of $^{13}\text{C}/^{15}\text{N}$ -labeled ATP and CTP. We have used this labeling technique in the lab and it works as expected. We do not show the data because we have yet to publish that work.

Reviewer #3 (Remarks to the Author):

The manuscript by Haslecker and co-workers presents an intelligent method for the labelling of nucleic acids based on an DNA polymerase mutant identified in the Holliger group a few years ago. The method allows to both introduce modified nucleotides at specific positions and to label with, for example, stable isotopes entire segment of the RNA. It has, potentially a broad range of applications in experiments looking at structure, dynamics and interactions of RNA. The work characterizes the activity of the polymerase in single and multistep reactions, which explores the range of possibilities and defines the experimental parameters.

The paper is overall well written and the figures are clear, although at times the information is not copious. The topic is relevant and the results interesting, this is in principle an elegant exploration of a very useful method.

Below are a number of points that needs to be addressed prior to publication.

Major point:

1- While the specificity of the reaction is generally clear, the yield, which is important to establish the potential of this strategy, is not really reported. The authors should report the final amount of RNA products per ml of reaction, for the different reactions they have performed.

Unlike regular transcription reactions, the yield is not directly related to the reaction volume, but can be calculated relative to input, that is the amount of segment 1. Typically, we start with ~5ml of 20 μ M segment 1, (~400 μ M of a 250 μ l NMR sample). Since the reaction proceeds to completion, we would expect a similar concentration for the extended product. To assess this, we have now included data on representative samples for which we simply wash the reaction six times with 5M Guanidinium-HCl to flush out unincorporated rNTPs and digested dNTPs. Indeed, we find close to 100% yields.

For structural biology, we expect that labs will use a more stringent purification protocol. We have provided yields for representative samples of one-step and two-step SegModTeX after gel purification and electro-elution. In our hands, there is about a 20% loss per purification cycle.

We have now included this information in Supplementary Table 1.

Minor points:

1- Purification of the TGK polymerase (page 5). Please describe in detail and show gels, chromatograms etc. This is important information to be able to use this method. This could be in the Supplemental materials.

The details for purification of TGK was included in the methods section and is a typical one-step purification due to the presence GST-tag. We have now included a gel in Supplementary Fig. 1a to show the purity after elution.

2- I would add a panel to Figures 1 where the actual experiment is represented in a cartoon form and the standard parameters are reported. This I think would help link the concept to the experimental conditions.

Done.

3- Page 5, one but last line. The bands of Seg1-seg2 in 2a are weak. It seems unlikely you one a seg1 band which is significantly weaker than seg1-seg2. Based on what I see on this gel, one can say seg1-seg2 is the only major species, but I am not sure how the authors can state with confidence that the turnover is 100%. How is this measured.

*The seg1-seg2 band are simply weaker because they have been diluted, a necessary step for the correct DNA concentration to get efficient DNase digestion (**Turbo DNase protocol**). We have tried to address this by stating the dilution factor in the well labels. We can be confident that the turnover is close to 100% because we do not see any leftover seg1 band. We have now also added a supplementary table that quantifies the yields and confirms this.*

4- Page 6 last line of the second paragraph and figure 2c. It seems to me G:dT runs as an extended oligo, i.e. as a 26mer.

As indicated in the cartoon in Fig. 2c, the seg1 itself is 1 base longer (26nt) compared to the other seg1s in the same gel (25-nt), so it is expected to run as a 26nt oligo. We have provided clarification of this in both the figure legend and in the text (lines 163-165).

5- Figure 2 – The legend for Figure 2d is missing

Done.

6- Figure 3, please add the spectrum of the fully labelled MLV, as a comparison.

Done. We have added the comparative fully labeled spectra, both in Fig. 3 and Supplementary Fig. 2.

7- Page 8, end of the first paragraph and Figure 4b. One of the adenine resonances, labelled with an asterisk, is not superimposing. Could you please explain and correct the statement in the paper as required.

We have changed the statement in the paper (line 210); indeed, one of the five resonances is not the exact match. We have been careful about using the same buffer, temperature, etc. for the experiment, our best explanation for that mismatch is the position of the labeled nucleotides is expected to be involved in transitions between two alternate conformations and thus possibly very sensitive to solution conditions as seen by the Tolbert laboratory.¹

8- Figure 5c, right. The spectrum of the AC and psi RNAs show what is probably noise at the frequency of U27. Could you please confirm and comment?

To address this question, we recollected the data with more scans to get a higher signal to noise ratio. Secondly, we now compare the sample with four replicates of tRNA^{lys3} produced by T7. This gives a good idea of the regions that are very sensitive to solution conditions, as seen in the new Fig. 5c and Supplementary Fig. 2 There is no noise at the U27 position.

9- It would be good to see repeats for the key reactions in the Supplemental methods.

This is now included in Supplementary Fig. 5.

References

- 1 Luo, L. *et al.* HnRNP A1/A2 Proteins Assemble onto 7SK snRNA via Context Dependent Interactions. *J Mol Biol* **433**, 166885, doi:10.1016/j.jmb.2021.166885 (2021).

REVIEWERS' COMMENTS

Reviewer #1 (Remarks to the Author):

In my view, the authors have done an excellent job revising the manuscript and responding to the reviewer queries and comments. I believe the revisions have substantially improved and clarified the manuscript. I recommend this work for publication with great enthusiasm and believe this innovative, timely method will be widely adopted by the community.

Reviewer #2 (Remarks to the Author):

The authors have adequately addressed my previous comments and suggestions and the manuscript is suitable for publication.

Reviewer #3 (Remarks to the Author):

The authors have addressed my queries. I look forward to seeing this elegant method published.